# HiSV: A control-free method for structural variation detection from Hi-C data

**Junping Li, Lin Gao** *, Yusen Ye

Department of Computer Science, School of Computer Science and Technology, Xidian University, Xi'an, Shaanxi, China

* lgao@mail.xidian.edu.cn

## Abstract

Structural variations (SVs) play an essential role in the evolution of human genomes and are associated with cancer genetics and rare disease. High-throughput chromosome capture (Hi-C) technology probed all genome-wide crosslinked chromatin to study the spatial architecture of chromosomes. Hi-C read pairs can span megabases, making the technology useful for detecting large-scale SVs. So far, the identification of SVs from Hi-C data is still in the early stages with only a few methods available. Therefore, we developed HiSV (**Hi**-C for **S**tructural **V**ariation), a control-free method for identifying large-scale SVs from a Hi-C sample. Inspired by the single image saliency detection model, HiSV constructed a saliency map of interaction frequencies and extracted saliency segments as large-scale SVs. By evaluating both simulated and real data, HiSV not only detected all variant types, but also achieved a higher level of accuracy and sensitivity than most existing methods. Moreover, our results on cancer cell lines showed that HiSV effectively detected eight complex SV events and identified two novel SVs of key factors associated with cancer development. Finally, we found that integrating the result of HiSV helped the WGS method to identify a total number of 94 novel SVs in two cancer cell lines.

## Author summary

Cancer and rare diseases are often driven by structural variations (SVs). Despite their importance, detecting SV events remains challenging. High-throughput chromosome capture (Hi-C) technology has proven valuable for large-scale SV detection. However, algorithms that can use Hi-C data without control samples for SV detection have been severely lacking. Therefore, we presented HiSV (**Hi**-C for **S**tructural **V**ariation), a control-free method for identifying large-scale SVs from a Hi-C sample. We evaluated HiSV's performance on the simulation datasets and cancer cell lines, HiSV achieved superior accuracy and sensitivity. Moreover, HiSV effectively captured complex SVs in cancer cell lines. Finally, we demonstrated that HiSV can be applied to supplement the result of WGS methods.

**Data Availability Statement:** Hi-C raw datasets of HCC1954, K562, T47D and MCF7 cell lines were downloaded from NCBI GEO database (accession: GSM3258551, GSM1551620, GSE63525, GSE105697 and GSE66733). WGS raw datasets of K562 and T47D were downloaded from GEO

database (accession: GSE176762 and
GSE176679). And total RNA-seq datasets for K562
and GM12878 were downloaded from GEO
database (accession: GSM5330997 and
GSM5330998, GSM5331122 and GSM5331123).
The source code of HiSV can be accessed at:
https://github.com/GaoLabXDU/HiSV.

**Funding:** This work was supported by the National
Natural Science Foundation of China [61873198 &
62132015 to L.G., 62002275 to Y.Y.]. The funders
had no role in study design, data collection and
analysis, decision to publish, or preparation of the
manuscript.

**Competing interests:** The authors have declared
that no competing interests exist.

## Introduction

Structural variations (SVs) are large-scale genomic alterations as a result of genetic events, such as deletions, duplications, inversions, and rearrangements, that involve 50 or more base pairs (bps) [1]. Numerous studies have shown that SVs play an essential role in the evolution of human genomes and are associated with cancers and other rare diseases [2,3]. Compared with single nucleotide variations (SNVs) and small indels, SVs are difficult to identify and have been less understood, even though they are more likely to have greater impacts on gene functions.

Microarray comparative genome hybridization (array CGH) and the short-read technology have been widely used to detect special types of SVs [4]. For example, Array CGH detects copy number variations (CNV) but has difficulties in detecting copy neutral variations such as balanced translocations and inversions [5]; short-read technology can identify SVs involving breakpoints with single base-pair resolution [6], but cannot be used to accurately detect SVs in the repetitive regions of the genome and complex SVs with multiple breakpoints [1]. Recent studies suggest that data obtained by the high throughput sequencing-based chromatin conformation capture (Hi-C) technique can be used to overcome limitations of the short-read technology to handle large-scale SVs [7]. As a relatively new technique, Hi-C is designed to study the spatial architecture of chromosomes by probing all possible genome-wide pairwise chromatin interactions, including interactions in the repetitive region of the genome that traditional short-read-based technology cannot identify. The possibility of using the Hi-C interaction matrix to detect a variety of SVs has been studied in [7]. Hi-C based SV detection methods such as HiCtrans [8], and HiNT-TL [9] are designed to detect inter-chromosomal translocations. HiC_breafinder [10] defined the SV breakpoints by searching the abnormal interaction block with higher interaction frequencies compared with the background model. More recently, the new method EagleC [11] combined deep-learning and ensemble-learning strategies to predict a full range of SVs. Both HiC_breakfinder and EagleC methods use several cell lines to construct a reference model that efficiently distinguishes intra-chromosomal SV signals from other chromatin interactions. HiC_breafinder [10] constructed a background model using several normal cell lines and defined the SV breakpoints by searching the abnormal interaction block with higher interaction frequencies compared with the background model. The new method EagleC [11] trained a multilabel classifier using several cancer cell lines and a GM12878 cell line to predict a full range of SVs. Both HiC_breakfinder and EagleC can efficiently distinguish intra-chromosomal SV signals from other chromatin interactions. However, a common obstacle in cancer genomes is the lack of an appropriate control sample for normal tissue [12]. Cell lines are a key tool in preclinical cancer research, but it is unclear to what extent they represent patient tumor samples [13], so the reference model constructed with cell lines often do not work well with patient samples.

Therefore, we described HiSV (**Hi**-C for **S**tructural **V**ariation), a control-free method based salient object detection model to identify large-scale SVs from a Hi-C sample. HiSV measured the saliency value of each pixel (bin pair) by calculating the dissimilarity between its interaction frequencies and that of neighboring pixels, and then segmented the saliency map by total variation regularization. The segments whose segmented saliency exceeds a certain threshold were chosen as the SV events. Based on simulation samples and cancer cell lines, HiSV achieved better performance than existing methods. Furthermore, long-read-based validation datasets demonstrated that HiSV is an effective tool for detecting complex SV. Moreover, HiSV identified novel SVs of key factors associated with cancer development by analyzing the differential expression genes and detecting neo-loops. Finally, by comparing and integrating the result of HiSV and WGS, we showed that HiSV can complement the incomplete identification of SVs from WGS methods.

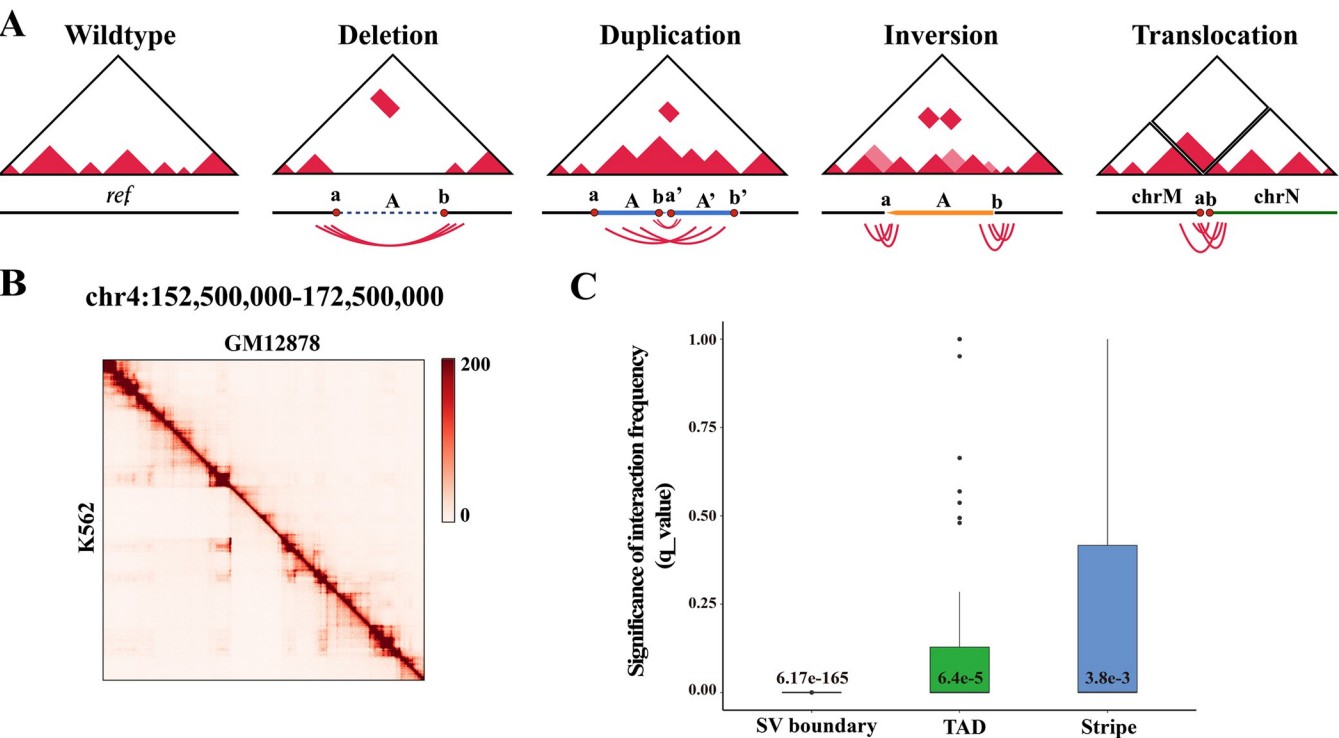

**Fig 1. Hi-C as a new technique for detecting large-scale SVs. (A)** The expected alteration to chromatin interaction frequencies for different types of SVs. **(B)** The Hi-C map is shown a validated deletion event in K562 cell line. **(C)** The significances of interaction frequencies of stripes, TADs and SV boundaries on chromosome 4 in K562 cell line.

## Results

### Overview of HiSV

In the Hi-C data, the interaction frequencies scales as a power law with genomic distance [14]. When SV occurs, the distally genomic regions are fused and the interaction in two loci around the breakpoint rises to the level of the proximal loci, causing marked deviations in interaction frequencies. As shown in Fig 1A, we draw Hi-C maps for different types of SV events. For example, a deletion of fragment *A* will result in the junction of breakpoint *a* and breakpoint *b*. Because of the spatial proximity, there will be strong interaction frequencies between breakpoint *a* and breakpoint *b*. When we map the Hi-C reads to the reference genome, a salience block was present at the breakpoints compared to the surrounding regions. Similarly, a duplication of fragment *A* will result in the junction of the original DNA fragment *A* and the duplication fragment *A*, resulting in a significant block between breakpoints *a* and *b* regions. For inversion, the sides of the breakpoint will form a 'butterfly shape' on the Hi-C map. For translocation, it will result in the junction of breakpoint *a* on chromosome *M* and breakpoint *b* on chromosome *N*, with unusually strong long-range trans interactions visible on the Hi-C maps between chromosome *M* and *N*.

At over megabase distances, the expected interaction frequencies between two loci close to zero, while abnormal interaction blocks formed by large-scale SVs are extremely prominent in the surrounding area. This level of significance extends well beyond other chromatin higher-order structures. For example, Fig 1B showed a 3.1Mb deletion event in K562 cell line. we could observe the unusually strong interactions at breakpoints compared with the normal cell line. We then compared the significance of the interaction frequencies of stripes, TADs and SV boundaries on chromosome 4 in the K562 cell line. The significances of interaction

frequencies were calculated by FitHiC2 [15]. FitHiC2 corrects the resulting binomial p-values for the multiple testing using Benjamini-Hochberg to compute q-values, which represent the minimum false discovery rate (FDR). TAD boundaries were identified by Insulation Score [16] and stripes were identified by StripeCaller [17]. As shown in the Fig 1C, we observed that the interaction frequencies formed by SV events are more significant than most TADs and stripes, with a median q-value of 6.17e-165 for SV boundaries where 6.4e-5 for TADs and 3.8e-3 for stripes. We then counted the significance of TADs large than 1Mb. The minimum q-value of these TADs is 4.59e-8, which is greater than the median q-value of the SV boundary. Overall, we demonstrated that large-scale intra-chromosomal SVs can be isolated from the Hi-C map. Therefore, we targeted the significant regions in the Hi-C matrix as large-scale SV events. Identifying large-scale SVs from a Hi-C map can be seen as an image salient object detection problem. The solution to this problem usually consisted of two stages: 1) detecting salient objects in the scene and 2) segmenting the entire range of these objects [18].

Here, we developed the HiSV method to identify large-scale SVs from a Hi-C sample based on saliency detection model (Fig 2). First, we computed the distance-normalized Hi-C interaction matrix to avoid the strong interactions on the diagonal interfering with the detection of SVs. The next step is to isolate the significant regions from complex background. To this end, we measured the saliency by calculating the local spatially weighted dissimilarity for each pixel (bin pair). Finally, we used the total variation segmentation to group the sparse salient subsets into several subgroups called segments, and segments will be reported as a SVs event if the segmented interaction frequencies greater than a predefined cutoff.

Our method supports raw fastq, bam file,.hic,.mcool and Hi-C Matrix files as its input. The final HiSV output file includes the regions and types for each SV event. More details of the HiSV are discussed in Materials and Method.

### HiSV outperforms existing methods in detecting SVs on Hi-C maps

We systematically evaluated the performance of HiSV by comparing it with existing methods, HiCtrans [8], HiNT-TL [9] and HiC_breakfinder [10] and EagleC [11]. We used simulation

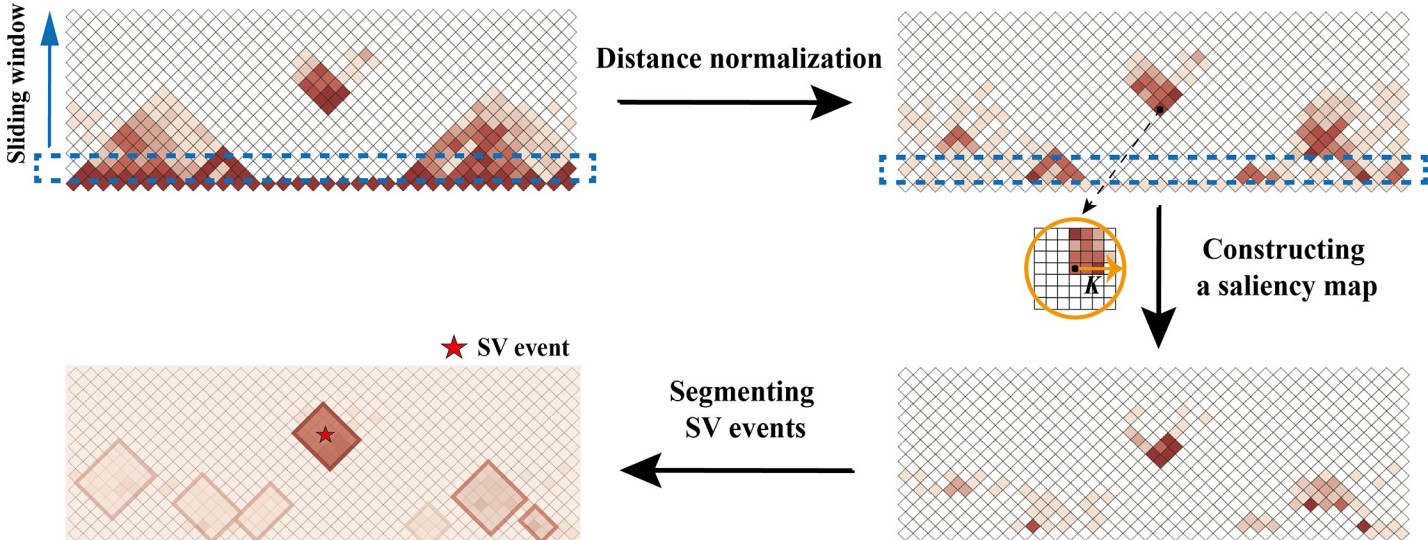

**Fig 2. The framework of HiSV.** HiSV has the three major steps: (1) Distance normalization: HiSV normalized the interactions based on linear genomic distance; (2) Saliency measure: the saliency value of each bin pair is measured by calculating the dissimilarity between its interactions and that of neighboring bin pairs; (3) Segmentation: HiSV used two-dimensional segmentation method to group the sparse data into several subgroups called segments, and a segment will be reported as a SVs event if the segmented interactions greater than a predefined cutoff.

data and four cancer cell lines, HCC1954, K562, T47D and MCF7 as the benchmark datasets. Here, we evaluated the performances of HiSV by comparing it with HiCtrans, HiNT-TL and HiC_breakfinder on HCC1954, K562 and T47D cell lines. Since K562 and T47D cell lines were used as training samples by EagleC, we then compared the performance of HiSV with EagleC on HCC1954 and MCF7 cell lines (Figs A and B in S1 File). To evaluate the impact of tumor heterogeneity on SV prediction, we simulated a series of Hi-C datasets containing multiple SVs at various sequencing depths and tumor purities (Materials and methods). Since HiCtrans and HiNT-TL can only detect inter-chromosomal translocations, we divided the result of SVs into two categories of intra- and inter-chromosomal SVs and compared them respectively. Additionally, the simulation data only contained chromosomes 10 and 11, while HiNT-TL does not accept single chromosome pair as input, so we excluded it when evaluating the performance using simulation data.

## Performance evaluating using simulation data

First, we evaluated the performance of HiSV and other methods at different sequencing depths (Fig 3A). For inter-chromosomal translocation, HiSV consistently has higher precision and recall rates than other methods at all sequencing depths. HiCtrans had the second highest recall for each sample. HiC_breakfinder does not support detection of inter-chromosomal translocations in the low coverage sample but had higher precision than HiCtrans at higher coverage. The F1-score of HiSV was significantly higher than other methods in detecting intra-chromosomal SVs. For example, the F1-scores of HiSV at 2X, 4X and 6X coverage were 45%, 55% and 57%, respectively, while HiC_breakfinder's F1-scores are 40%, 40% and 47%.

We further measured the F1-score of these methods at varying tumor purities (Fig C in S1 File). We observed that HiSV achieved higher F1-scores at all tumor purity levels for inter-chromosomal translocations. Especially, HiSV's F1-score was most dramatic in low tumor purity samples (< 0.4 tumor purity). For example, the F1-score of HiSV was 67% at 0.2 tumor purity, while the HiCtrans and HiC_breakfinder were 8% and 23%, respectively. For intra-chromosomal SVs, the F1-score of HiC_breakfinder was 6 percentage points higher than that of HiSV in the 0.8 tumor purity sample, while the F1-score of HiSV was higher than that of HiC_ breakfinder in all other tumor purity samples.

**Performance evaluating using cancer cell lines.** We then evaluated the performances of HiSV by comparing it with existing methods on three tumor cell lines. Different methods vary considerably in precision, recall rate and AUPR. By allowing 100 K and 1 Mb mismatches on both sides of the breakpoint, HiSV outperforms all other methods and achieves higher F1 scores and AUPR for inter-chromosomal translocations in HCC1954 and T47D cell lines. (Figs 3B and D in S1 File). HiSV's superior F1-score was most dramatic in HCC1954 cell line, which was 14% and 28% higher than the second-ranked HiC_breakfinder (0.31 VS 0.17, 0.6 VS 0.32) when allowing mismatch is 100Kb and 1Mb, respectively. HiNT-TL had the third-highest F1-score in all three cell lines, followed by HiCtrans. For intra-chromosomal SVs, HiSV identified 171 SVs in HCC1954, and 23 SVs in K562, while T47D reported 17. HiC_-breakfinder detected 131, 12 and 13 SVs in three cancer cell lines respectively. Notably, by allowing 100 Kb mismatches, HiSV had the higher F1-score and AUPR in HCC1954 and K562 cell lines (Figs 3B and D in S1 File). However, when we reduced the allowed mismatch from 100K to 50Kb, HiC_breakfinder surpassed HiSV in F1-score as it effectively optimized the initial 100Kb resolution breakpoint to 10Kb resolution. When we polish the breakpoint of HiSV to 10Kb resolution as well, HiSV regained its superiority and achieved higher F1-scores in all three cell lines (Fig K in S1 File). The higher precision of HiC_breakfinder came with significant sacrifice on recall. However, HiSV showed weaker performance than HiC_breakfinder in

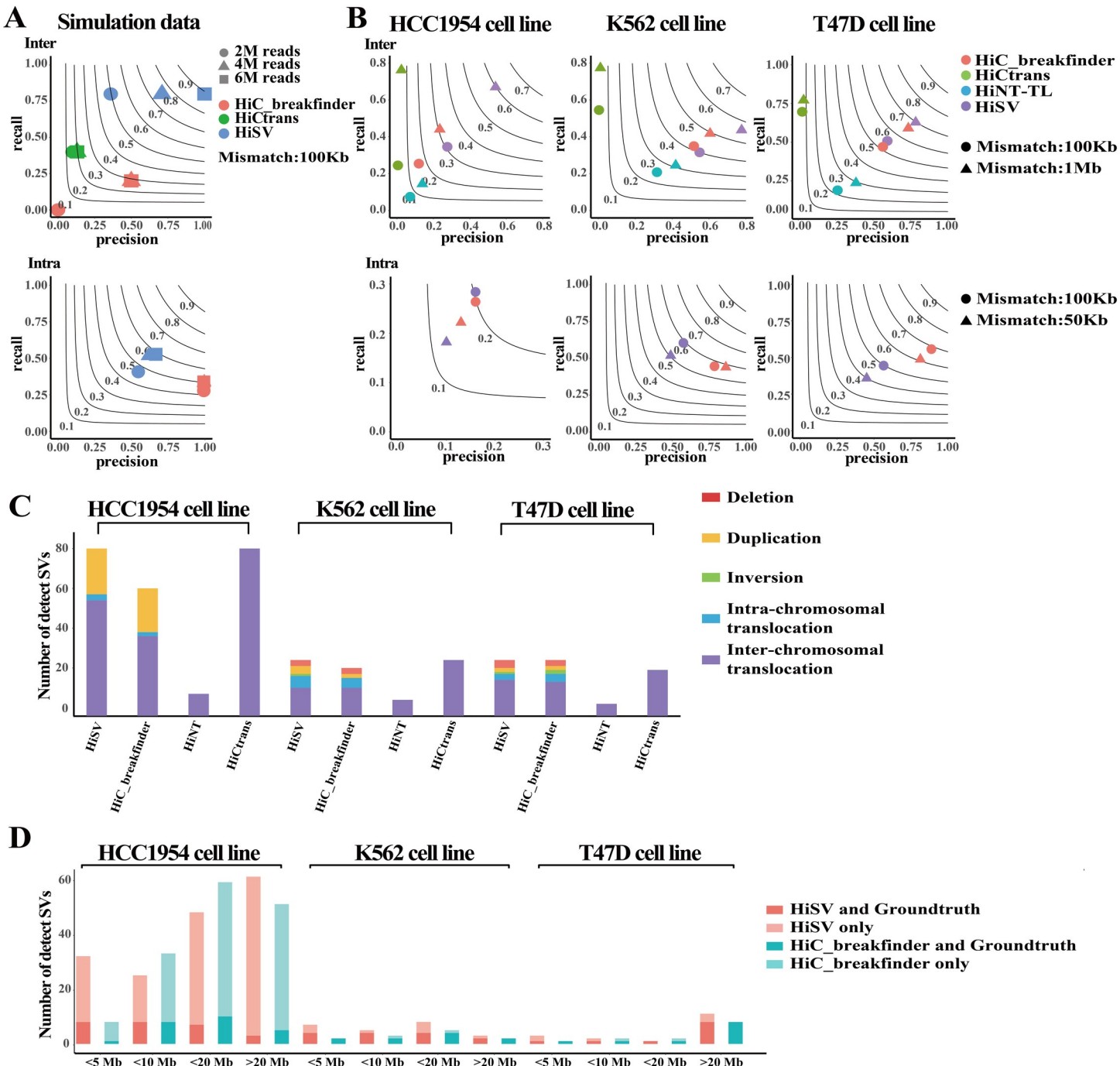

**Fig 3. Performance comparison of SVs callers. (A)** The precision and recall rates of SVs detected by HiSV, HiC_breakfinder, HiCtrans and HiNT-TL in the simulation sample at different sequencing depths. Black contours show harmonic means of precision and recall rates (F1-score). **(B)** The precise and recall rates of SVs detected by the above methods in HCC1954, K562 and T47D cell lines. **(C)** The number of validated SVs with different types detected by existing methods. **(D)** The number of different ranges of SVs detected by HiSV and HiC_breakfinder with validation by groundtruth.

detecting intra-chromosomal SVs. One important factor contributing to this result could be that HiC_breakfinder was involved in the creation of the true set of SVs, which may provide it with an inherent advantage in this evaluation.

We performed more in-depth comparisons between HiSV and HiC_breakfinder in different SV types. First, we showed the number of different types of validated SVs detected by these methods in Fig 3C. We can observe that HiSV detected not only the highest number of SVs but also all variant types. Then, we evaluated the performance of different SV types (Fig E in S1 File). We found that HiSV has a consistently higher F1-score than HiC_breakfinder across nearly all SV types. For example, the F1-scores for HiSV to detect duplication in HCC1954 and K562 cell lines were 20% and 80% respectively, while the F1-scores for HiC_breakfinder were 19% and 50%.

Furthermore, we compared the result of HiSV and HiC_breakfinder for intra-chromosomal SVs at different SV sizes. We then divided the SVs detected by each method into four categories according to their length, namely <5Mb SVs, <10Mb SVs, <20Mb SVs and >20Mb SVs. As shown in Fig 3D, we found HiSV and HiC_breakfinder detected similar numbers of >5Mb SVs, whereas HiSV detected more <5Mb SVs in all three cell lines.

## HiSV detects complex SVs in cancer cell lines

We found that the SVs detected by HiSV contained several complex SV events which are a chain of simple SVs and have multiple breakpoint connections. These complex SVs were validated by long-read sequencing data. HiSV detected four complex SVs in K562 and T47D, and these complex SVs event were not fully detected by HiC_breakfinder. For example, a large-scale complex SV event in K562 span ~10Mb, involving two overlapping intra-chromosome translocations t(18) (p11.32;p11.31) and t(18) (p11.31;p11.22) (Fig 4A). And on chromosome 10 of T47D, complex SV span ~30M, involving two overlapping duplications dup(10) (q22q24) and dup(10) (q23q24)) (Fig 4B). Furthermore, the complex inter-chromosomal rearrangement event (chr3-chr10) in T47D involved five fragments with five breakpoints (Fig 4C). And another complex inter-chromosomal rearrangement event in K562 crosses three chromosomes t(9;13;22) involving multiple duplications and multiple inter-chromosomal translocations. This complex SV involves BCR/ABL1 gene fusion, which is a hallmark of K562 (Fig 4D) [19]. Based on the above observation, HiSV can be an effective complex SV detection tool.

## Effects of SVs on gene expression

We next sought to investigate the effect of SVs on gene expression. As shown in Fig 5A, a total of 2569 differentially expressed genes (DEGs) were detected between GM12878 and K562 cell lines. We found the 194 DEGs associated with SVs (Fig 5B). For example, FNBP1 gene expression was significantly downregulated because of expression disruption by duplication dup(9) (q12) and GNB1L gene was upregulated undergoing inversion inv(22) (q11.21).

In addition, we found novel SVs are associated with 47 DEGs (Fig 5C). To further explain the mechanism of the gene mis-regulation, we used Neoloopfinder [20] to reconstruct the Hi-C matrix at the SV loci and detect neo-loop formed by SVs, which may correspond to regulatory event of genes [21]. Fig 5D and 5E illustrated that both GPC5 and DLGAP1 are the neo-loop involved genes, in which DLGAP1 gene affects the growth rate of hematopoietic cells [22] and GPC5 is a promising therapeutic target for reducing podocyte vulnerability in glomerular disease [23]. Interestingly, we found DLGAP1 gene with an enhancer hijacking event because it is located in one anchor of the neo-loop, and an enhancer located in the other anchor of the neo-loop. In conclusion, HiSV is a sensible method for disease-associated SV discovery and can be used as a preliminary tool to explain the remote regulation of gene expression caused by SVs.

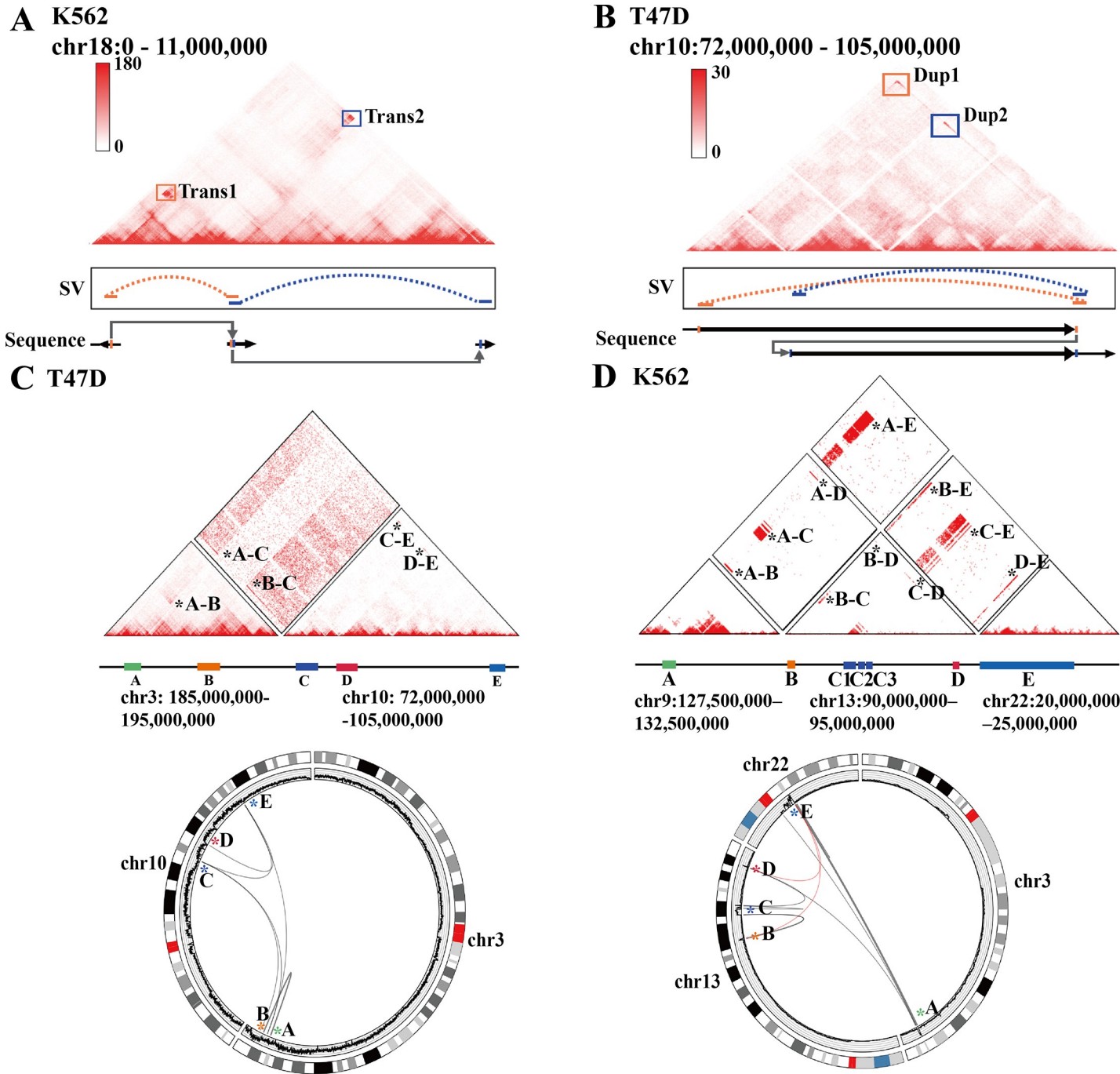

**Fig 4. Complex SVs were detected by HiSV.** Intra-chromosomal complex SVs were detected in K562 (A) and T47D (B) by HiSV. The top shows the Hi-C maps with complex SVs and boxes represent the SVs event detected by HiSV. The plot in the middle depicts breakpoint connections that are detected from long-read sequencing data. The bottom panel shows the genome structure of the complex SVs. Inter-chromosomal complex SVs were detected in T47D (C) and K562 (D) by HiSV. The top panel shows Hi-C maps with complex SVs. The circos plot in the below, from outside to inside, depicts the chromosome ideogram, copy number profiles and breakpoint connections. The black arcs are the SVs detected from long reads data and red arcs represent the inter-chromosomal rearrangements detected by all Hi-C methods.

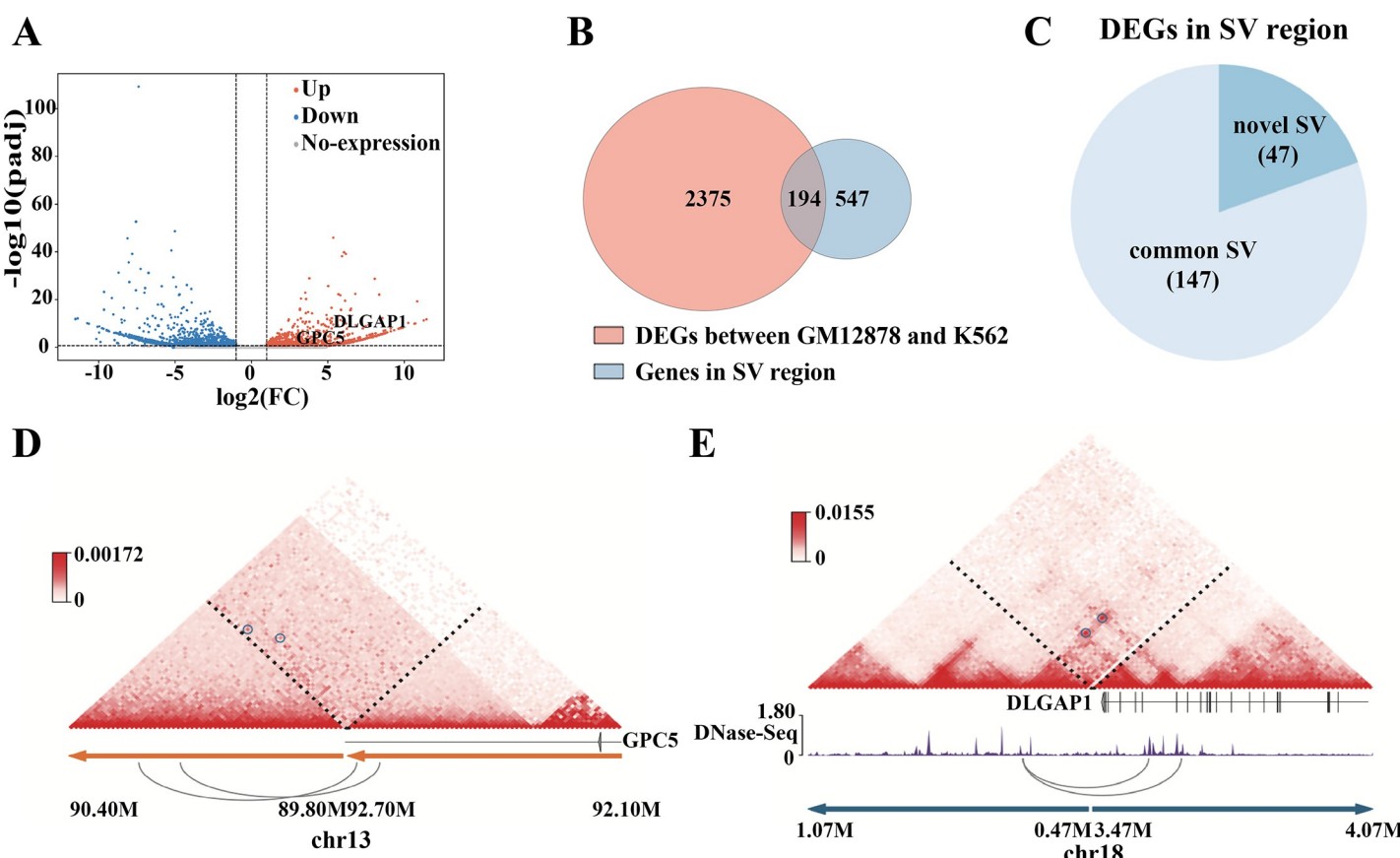

**Fig 5. Effects of SVs on gene expression in K562 cell line.** (A) Volcano plot showing differentially expressed genes (DEGs); red dots represent upregulated genes; blue dots represent downregulated genes and gray dots represent genes with no expression changes. (B) Comparison of the genes in SV regions with DEGs. (C) The overlap counts of DEGs in novel SV regions and common SV regions. (D) The differential expression of GPC5 is associated with neo-loops. (E) The upregulated expression of DLGAP1 is associated with an enhancer hijacking event.

## HiSV complements the detection of SVs by the WGS method

**Comparison of SVs detected by HiSV and WGS methods.** To analyze the performance of different sequencing technologies for detecting SVs, we compared the ability of HiSV and WGS methods for detecting SVs in K562 and T47D. Each type of SVs was divided into different categories according to the size (DEL<1Mb; DEL>1Mb; DUP<1Mb; DUP>1Mb; INV<1Mb; INV>1Mb). We counted the results of WGS methods and HiSV for each category of SVs (Fig F in S1 File). We observed that HiSV can effectively detect more large-scale SVs, particularly complex SVs (Fig 6A). On the contrary, WGS methods provided higher detection capabilities for small-scale SVs. These results are consistent with the previous study [24]. Overall, Hi-C and WGS have unique strengths for identifying SVs. Therefore, integrating WGS and Hi-C can comprehensively detect SVs.

**Integration of SVs detected by HiSV and Lumpy.** We showed the integrated result of SVs for K562 and T47D in Figs 6B and G in S1 File. For K562 cell line, we obtained 8,243 calls including 5330 deletions, 1184 tandem duplications, 70 inversions, 1407 intra-chromosomal translocations, and 252 inter-chromosomal translocations. For T47D cell line, we obtained 6,317 calls including 4038 deletions, 1062 tandem duplications, 67 inversions, 904 intra-chromosomal translocations, and 246 inter-chromosomal translocations. Notably, compared with the WGS result detected by Lumpy, the supplementary HiSV results allow 44 and 50 novel SVs

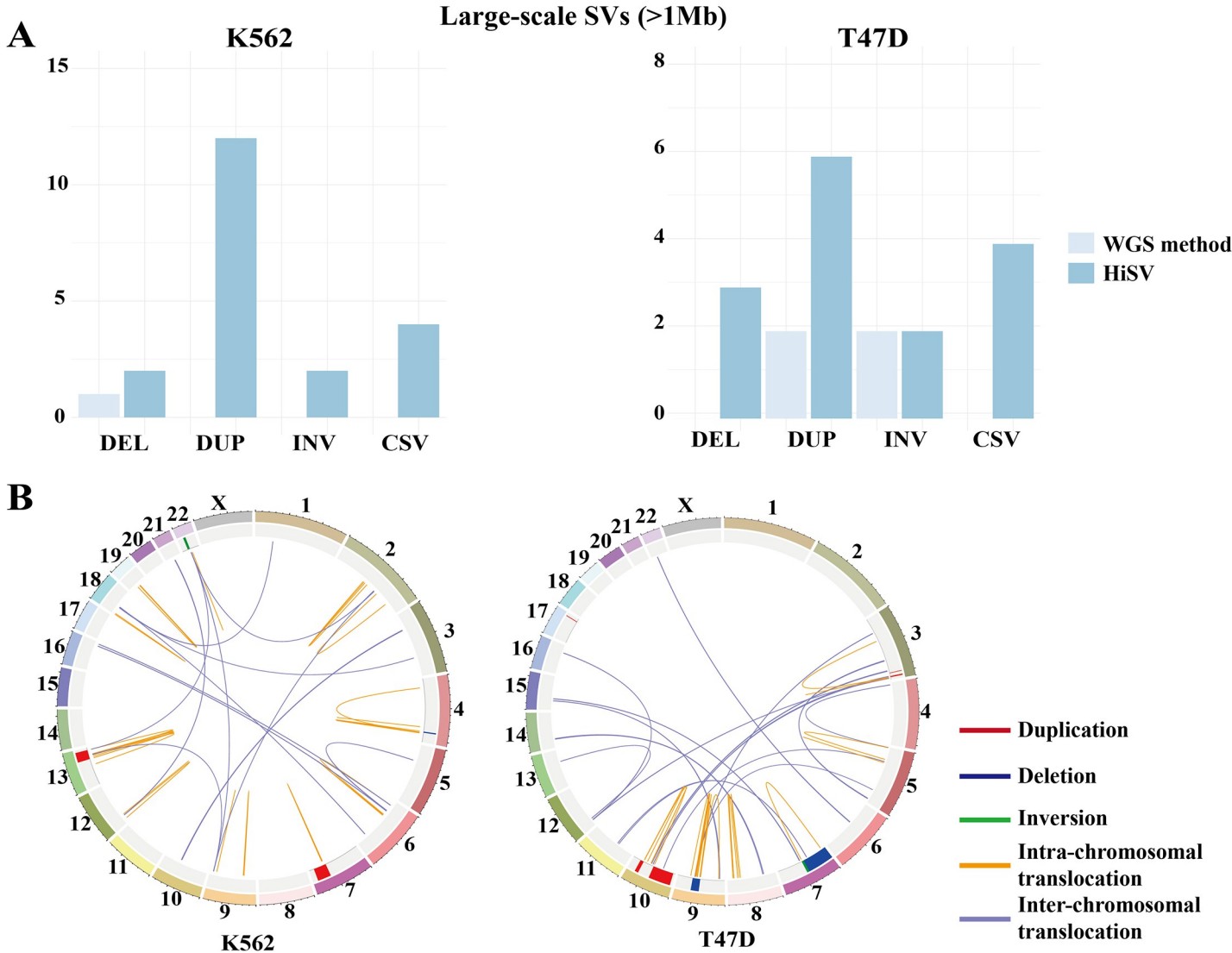

**Fig 6. HiSV complements the detection of SVs by the WGS method. (A)** The detection results of WGS method and HiSV for different types of large-scale SVs in K562 (left) and T47D (right) cell lines. **(B)** Circos plot visualizing the detection of novel SVs in K562 (left) and T47D (right). Tracks from outer to inner circles are the chromosome ideogram, duplication (red) and deletion (blue), and positional rearrangements including inversions (green), intra-chromosomal translocation (orange), and inter-chromosomal translocation (purple).

to be detected for K562 and T47D cell lines respectively. These novel SVs mainly include large-scale SVs and translocations. To sum up, this model takes the result of HiSV as a supplement to WGS methods which improves the performance of detecting SVs. Therefore, HiSV can complement the incomplete identification of SVs from WGS method.

## Discussion

Hi-C provides an excellent opportunity to simultaneously identify large-scale SVs and study the regulation of disease-associated gene expression. Because Hi-C data provides longer-range information than standard short-read, so it breaks the limitation of short-read-based WGS in detecting SVs. However, computational tools that detect a broad range of SVs from a single Hi-C sample are still lacking. Here, we described HiSV (**Hi**-C for **S**tructural **V**ariation), a

computational pipeline based salient object detection model to identify large-scale SVs from a single Hi-C sample. HiSV measured the saliency value of each pixel (bin pair) by calculating the dissimilarity between its interaction frequencies and that of neighboring pixels, and then segmented the saliency map by total variation regularization. The segments whose segmented saliency exceeds a certain threshold are chosen as the SV events. HiSV supported multiple input formats including raw FASTQ, BAM and contact matrix. The output file of HiSV included the position and type of SVs. We applied HiSV to simulated and real data, and our analysis highlighted some of the challenges of detecting SVs using Hi-C data.

Based on the evaluations both of simulated and real data, HiSV achieved a higher level of accuracy and sensitivity compared to existing methods. Especially, HiSV didn't require multiple normal samples to construct a reference model. It not only reduced the cost of the experiment but also avoided the influence of heterogeneity among samples on the result. However, HiSV has limited power in detecting SVs in < 1Mb in size. There are two main reasons: 1) the resolution is limited because Hi-C relies on the presence of digestion sites kilobases apart in the genome. 2) It is difficult to distinguish the increased interactions whether due to small-scale SVs or normal variation in 3D genome organization. To be able to solve these problems, we need have a deeper understanding of the 3D genome. Moreover, we used K562 and T47D cell lines with long-read sequencing, RNA-seq and DNase-seq demonstrated HiSV effectively identified the complex SV events and novel SVs of key factors associated with cancer development.

We observed that HiSV is a powerful tool to detect large-scale SVs. On the other hand, WGS is successful in detecting small-scale SVs with the highest resolution. Therefore, integrating the Hi-C and WGS can capture more SVs. However, both Hi-C and WGS depend on short-read sequencing, the problem of a high alignment error cannot be avoided. Therefore, full-spectrum SV detection requires the integration of more technologies.

## Materials and methods

### Implementation of HiSV

**Distance-normalized Hi-C matrix.**   In our method HiSV, interaction frequencies in the Hi-C matrix are scaled and normalized to weaken the frequency signals between genomic loci. Specifically, the normalized interaction frequency between bin $i$ and bin $j$ is a z-score defined as follows:

$$z_{i,j} = \frac{a_{i,j} - \mu(d)}{\sigma(d)}$$

where $a_{i,j}$ denotes the interaction frequencies between bin $i$ and bin $j$, $\mu(d)$ and $\sigma(d)$ are the mean and the standard deviation of the interaction frequencies of loci bins within distance $d = |j-i|$.

**Constructing a saliency map of interaction frequencies.**   When we measured the saliency, two factors were considered: the dissimilarity of the interaction frequencies and their spatial distance. With the increasing of the spatial distance between two bin pairs, the influence of the dissimilarity between them was decreasing. The saliency is normalized to range [0, 1] and defined as:

$$b_{i,j} = 1 - exp\left(-\frac{1}{(2k)^2}\sum_{p=i-k}^{i+k}\sum_{q=j-k}^{j+k} d[(i,j),(p,q)]\right)$$

where $k$ is the local region window. $d[(i,j),(p,q)]$ is the spatially weighted dissimilarity

measure between the element $(i, j)$ and element $(p, q)$ and defined as:

$$d[(i,j),(p,q)] = \frac{z_{i,j} - z_{p,q}}{1 + dist[(i,j),(p,q)]}$$

Where $p \in (i-k, i+k)$ and $q \in (j-k, j+k)$. $dist[(i,j),(p,q)]$ is the Euclidean distance between the element $(i, j)$ and element $(p, q)$.

**Segmenting SV events from the saliency map.** Here we used total variation regularization to fit the saliency profile and extract the salient segments as SV events [25]. The output of the total variation regularization is obtained by minimizing a particular cost function. Given a saliency map, the goal is to estimate the segmented data $B'$ which minimized the:

$$\min_{D'} \ E(B, B') + \lambda V(B')$$

where the first term is the $L2$ norm between the $b_{i,j}$ and the segmented value $b'_{i,j}$. The second term is the total variation penalty and $\lambda$ is regularization parameter. The total variation penalty was defined as:

$$V(B') = \sum_{i,j} \sqrt{|b'_{i+1,j} - b'_{i,j}|^2 + |b'_{i,j+1} - b'_{i,j}|^2}$$

Finally, we return the segments whose segmented saliency exceed threshold $t$ as the SV event.

## Determination of SV breakpoints

We first find SVs at a given bin size (eg.50Kb). After this initial phase, we further analyze the submatrix near the SV breakpoints with smaller binsize (eg.1Kb). As shown in Fig H in S1 File, the breakpoints of SVs can be recognized by locating the change point in the submatrix. We performed PCA on the rows and columns of the matrix separately, and breakpoints were determined where the sign of the first eigenvector or principal component changes [20].

## Annotation of SV types

The most basic way to represent Hi-C data is in matrix format. To make the method more general, we infer the type of SV event by Hi-C matrix instead of the bam file. Here, we analyzed the different patterns and signatures of structural variations across Hi-C contact matrix and classified the result into duplication, deletion, inversion, intra-chromosomal translocation, and inter-chromosomal translocation. Among them, translocations are divided into unbalanced and balanced translocations. Due to HiSV providing two breakpoint regions for each SV (the left fragment A and the right fragment B), we used the start position of fragment A and the end position of fragment B as the breakpoint of each SV.

Firstly, SVs were divided into inter-chromosomal translocation and other types based on whether the chromosome of the source locus is the same as that of the target locus. Then, we divided other types of SVs into copy number changed SVs and copy neutral SVs. Here, we defined the average coverage profile across the genome $c'$, which is positively correlated with copy number [9]. First, we computed a one-dimensional coverage profile at each bin across the genome. We then employed a generalized additive model [14] on the coverage profile to correct the bias in GC content, mappability, and the number of restriction sites. A structural variation will be defined as deletion if its average coverage profile is less than 90% of $c'$. On the contrary, the average coverage profile of the region is more than 110% of $c'$ and is determined as duplication. For other types of SVs, the average coverage profile between two breakpoints

floats around with $c'$, but there are different interaction patterns between the break-point regions. As shown in Fig I in S1 Fig, unbalanced trans-location forms a single block with strong interaction frequencies and balanced translocation and inversion both showed interaction frequencies split between two blocks, producing a 'butterfly', but in different orientations. Therefore, we distinguish inversion, balanced translocation, or unbalanced translocation by measuring in which the direction of the interaction frequency decreases in the breakpoint region. A structural variation will be defined as balanced translocation if the direction in which the interaction frequency decreases is constant. When the direction in which the interaction frequency decreases of structural variation breakpoint region change from top-down to bottom-up, it is determined as a balanced translocation. Instead, it is defined as inversion.

## Integrating SV signals from Hi-C and WGS

By comparing the SV calls from HiSV and WGS methods, we demonstrated that Hi-C and WGS have unique strengths for identifying SVs. Therefore, integrating Hi-C and WGS can comprehensively identify SVs. In general, simple integration used overlap to merge SV calls from different technologies. Although it achieved high precision detection of SV, the unique SVs detected by each technology are missed. Therefore, we integrated SV detection signals from Hi-C and WGS to comprehensively identify SVs. Here, we used Lumpy to integrate the multiple SV signals, which is a probabilistic framework for detecting SVs and is readily extensible to new signals from new technologies. Firstly, since the breakpoint may appear in the previous bin or the next bin, we expanded a bin at both ends of the breakpoint detected by HiSV and put them into the model as prior knowledge. Then, paired-end sequencing and split read sequencing were extracted from WGS data by BWA-MEM and the probabilities of read-pair module and split-read module were calculated by Lumpy. Finally, overlapping breakpoint intervals were clustered and the probabilities are integrated. The breakpoint can be determined by the position with probabilities in the higher percentile of the distribution. For Lumpy, the tuning parameters min_non_overlap was set to 150, weight set to 1 and min_mapping_threshold was set to 1.

## Simulating Hi-C data with SVs

To provide a benchmark for HiSV and other methods calls, we developed a simulation pipeline for generating Hi-C data with SVs. The simulated Hi-C data should not only contain a full range of SV events but also had chromatin higher-order structures. FreeHi-C [26] was recently developed to simulate high-fidelity to biological Hi-C data. Here, we extend FreeHi-C with embedded SV events. The methods are described in detail below.

We first used Hi-C data from a normal cell line to train the frequency of each fragment interactions by FreeHi-C and learn the expected contact on genomic distance using a smooth spline fit. We then generated a set of SV events and mapped them to RE sites. The distance between each RE fragment is recalculated based on SV events and obtained the expected interactions among the pairs of RE sites using their new genomic distance. We defined the initially expected interactions between each RE fragment as $f_1$ and the new expected interactions as $f_2$. We scaled the interactions of each fragment according to $\frac{f_1}{f_2}$. Subsequently, we generated pairs of sequencing reads from the new interaction fragment pairs by FreeHi-C.

Here, we chose chromosome 10 and chromosome 11 from hg38 as the reference and GM12878 as the training sample. To evaluate the effect of sample heterogeneous, we consider two factors to configure the simulation including sequencing depth and tumor purity. First, we simulated samples with 2M (~1X coverage), 4M (~2X coverage) and 6M (~3X coverage) read pairs respectively. Then, for the 6M read pairs sample, we mixed samples containing SVs

with normal samples in the proportions of 0.2, 0.4, 0.6, and 0.8, respectively. Each sample contains seven types of SVs including deletion, duplication, inversion, intra-chromosomal translocation, inter-chromosomal translocation, and three complex SVs, and the length of SVs from 1Mb to 2Mb (S1 Table).

## Evaluation of SV callsets

The true set of SVs for HCC1954 was obtained from ICGC PCAWG project (https://dcc.icgc.org/releases/PCAWG/cell_lines/HCC1954). The high confidence SVs for K562 and T47D were obtained from [10], and they were defined as detected on at least two of the three different platforms (Optical mapping, Hi-C and WGS), where the result of Hi-C is detected by HiC_breakfinder. The complex SV callsets of long reads data for T47D and K562 were detected by long-span paired-end tag (PET) [27] and linked-read sequencing [28]. We defined the SVs with lengths larger than 1Mb from these data sets are used as groundtruth.

The SV calls were assessed based on the groundtruth in the following approach. When the gap between the breakpoint and ground truth is within the allowed mismatch, the prediction SV is classified as true-positive. For a more robust evaluation that covers the resolution set by each method, we allow 50K and 100Kb mismatches for intra-chromosomal SVs, 100 K and 1 Mb mismatches for inter-chromosomal SVs.

## Hi-C and WGS data processing

For raw Hi-C sequencing data, BWA-MEM [29] is used to align read pairs to the hg38 reference genome. After the SAM files have been generated, we used HiCExplore [30] to generate a normalized Hi-C matrix, the resolution is set to 50kb. For raw WGS sequencing data, we used BWA-MEM [29] to create SAM files and used SAMtools [31] to generate BAM files.

## SV detection from WGS data

SV detection from WGS was carried out using Manta [32] and Lumpy [33]. Default parameters were used to run Manta and Lumpy. Control-FREEC [12] was used to detect CNV from WGS data, and we used $binsize = 1000bp$ to determine the breakpoint in K562 and T47D cell lines. We integrated the result of the above three methods as the result of WGS, where SV events are defined as detected on at least two of the three different methods.

## Calling the DEGs from RNA-seq

Since SVs not only affect gene dosage but also affect the regulatory elements which can regulate genes by long-range chromatin interactions, we expanded the results of SVs by 1Mb to find genes associated with SVs. Here, we used DEseq2 [34] to identify differentially expressed genes (DEGs) from RNA-seq dataset of GM12878 (lymphoblastoid cell line) and K562 cell lines. DEGs with false discovery rate (FDR) < 0.05 and absolute Log2 fold change (Log2 FC) > 1 were identified.

## Determination of the HiSV parameter

There are three parameters in HiSV: 1) the local region window $k$. 2) the regularization parameter $\lambda$. 3) the threshold $t$ for filtering SV segments. HiSV required a single parameter $k$ to control the local region window in measuring saliency, and the range of the parameter is associated with the resolution of Hi-C matrix. Previous studies [35] have shown that the average size of 3D genome organization was less than 1Mb. Therefore, the local region is approximately 1Mb which can avoid the normal 3D organization having high salience. Since we chose the binsize is 50 kb for Hi-C data, the choice of $k$ is 10. For the regularization parameter $\lambda$, we

chose 0.2 because it allows the segment to capture the complete SV region by analyzing the result of multiple experiments. HiSV filtered segments as SV events depend on one parameter, and the range of the parameter is associated with the sequencing depth of Hi-C data (Fig J in S1 File). When the Hi-C data with higher sequencing depth, the domain structures are more complete and the segments with higher saliency are defined as SV events. By evaluating the performance of HiSV at different sequencing with different parameter t, we have shown that for Hi-C data with a sequencing depth of nearly 100 million contacts, t is set to 0.6, and when the sequencing depth is less than 100 million contacts, t is set to 0.5 to achieve better results (Fig J). Here, for the cell lines with low sequencing depth (K562, MCF7 and HCC1954), we chose the parameter $t$ is 0.5, and for the cell lines with high sequencing depth (T47D), we chose the parameter $t$ is 0.6.

## Supporting information

**S1 File. Supplementary File. Fig A.** The precise and recall rates of SVs detected by HiSV and EagleC in MCF7 and HCC1954 cell line. **Fig B.** Size distribution of validated intra-chromosomal SVs detected by HiSV, HiC_breakfinder and EagleC. **Fig C. Performance comparison of HiSV with existing methods in simulation samples.** The F1-score is used to evaluate the sensitivity of HiSV and other methods to detect SVs in different tumor purity samples. **Fig D. Performance comparison of HiSV with existing methods in cancer cell lines.** The AUPR is used to evaluate the result of HiSV and other methods to detect SVs in different samples. **Fig E.** Performance comparison of HiSV and HiC_breakfinder in different SV types. **Fig F. Comparison of SVs detected by HiSV and WGS methods.** The detection results of WGS methods and HiSV for different types of SVs in K562 (a) and T47D (b). **Fig G.** The detection results of integrating WGS and Hi-C for different types of SVs. **Fig H. Determination of SV breakpoints.** The breakpoint was determined by searching the sign of the first eigenvector or principal component changes. **Fig I. Classification of the different types of SVs.** The cartoon in the box depicts the direction in which the interaction frequently decreases within the breakpoint region. **Fig J. Suggested $t$ of HiSV.** We assessed the effect of bin size (a), sequencing depth (b) and assay (c) on the choice of parameter $t$. **Fig K.** The precise and recall rates of SVs detected at high resolution by HiSV, HiC_breakfinder and EagleC in MCF7 and HCC1954 cell line. (DOCX)

**S2 File. Detailed information of datasets.**
(XLSX)

**S3 File. True positive dataset for each cancer cell line.**
(XLSX)

**S4 File. SV events detected in each cancer cell line.** Revised results for SV events by HiSV in each cancer cell line after code revision and additional analyses to compare HiSV with other tools.
(XLSX)

**S1 Table. The number of different types of SVs in simulation datasets.**
(XLSX)

## Acknowledgments

The authors thank Yong Gao Prof. for writing improvement and discussion. They also thank all the members of Prof. Gao's lab for their helpful suggestions.

## Author Contributions

**Methodology:** Junping Li, Lin Gao, Yusen Ye.

**Software:** Junping Li.

**Writing – original draft:** Junping Li.

**Writing – review & editing:** Lin Gao, Yusen Ye.

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
