## [Decision Letter · Decision Letter 0]

10 Sep 2022

Dear Dr. Gao,

Thank you very much for submitting your manuscript "HiSV: a control-free method for structural variation detection from Hi-C data" for consideration at PLOS Computational Biology.

As with all papers reviewed by the journal, your manuscript was reviewed by members of the editorial board and by several independent reviewers. In light of the reviews (below this email), we would like to invite the resubmission of a significantly-revised version that takes into account the reviewers' comments.

Especially, some of the reviewers raised serious concerns about this work.

We cannot make any decision about publication until we have seen the revised manuscript and your response to the reviewers' comments. Your revised manuscript is also likely to be sent to reviewers for further evaluation.

Sincerely,

Jie Liu

Academic Editor

PLOS Computational Biology

Jian Ma

Section Editor

PLOS Computational Biology

Reviewer's Responses to Questions

**Comments to the Authors:**

Reviewer #1: The authors present HiSV (Hi-C for Structural Variation), an algorithm to find structural variants from Hi-C contact maps.

The advantages of HiSV include:

- it is control-free and works for more tissues or primary samples,

- it detects CNV, intra/inter-chrom translocations at the same time with a decision tree.

However, the evidence and experiments in this paper are not sufficient to prove HiSV is reliable. My detailed comments are as follows:

1. There is not enough evidence to demonstrate that intra-chromosomal SVs can be separated from normal Hi-C contacts. Fig 1C shows that SV-related interactions are generally stronger than TADs. However, to make sure SVs can be separated from other structures:

1) SVs should be compared with the strongest TADs (Does “top 20 q_values” mean this in the figure caption? The authors should clarify.),

2) contact hot spots are not necessarily TADs, loops/stripes should also be considered.

2. A predefined threshold is used to select SVs from other segments. How is the threshold chosen? Should it be dependent on map resolution, sequencing depth, or assay (Hi-C/capture Hi-C/Micro-C/…)? The authors should provide more information.

3. Another way to evaluate the performance of the “predefined threshold” is to change it and calculate the AUROC. If this is also possible for other baseline methods, AUROC provides a more reliable comparison.

4. In Fig 3 (a and c), the precision and recall are calculated with all SVs. Is it possible to evaluate the performance in different SV types? If HiSV performs well in CNV, it also helps answer my questions in Comment 1.

5. From Fig 3c, I can see many false positive SVs called by HiSV, which might also be related to the limited separation I mentioned in Comment 1. What will we get if applying HiSV to a “normal cell”?

6. The authors should proofread the paper to make sure concepts are well explained in the text and figure captions. For example,

1) the “top 20 q_values” in Comment 1,

2) “DEG” or “DGE”? The full name should be provided when the abbreviation is mentioned for the first time,

3) I think the “dist()” part in formula 3 is wrong.

4) the colors of the two types of translocations in Fig 6b are too similar to distinguish.

7. Is it true that only “above threshold” pixels are chosen? Is it possible that some depleted contacts also correspond to SVs?

8. In Line 88, “the interaction frequencies between two loci decrease logarithmically with genome distance”. I think people usually use “power law” to describe the distance decay of contacts.

9. Can HiSV support other common Hi-C formats like .hic or .mcool?

Reviewer #2: In this manuscript, Li et. al. present HiSV, a saliency-based method, which aims to identify structural variants from HiC data. Compared to other SV detection tools based on HiC, HiSV is control-free and achieved better performance on simulated data and cancer cell lines. In addition, HiSV can complement the SV identification from WGS methods. The manuscript is clearly written and well organized.

I have four major concerns.

(1) Calling the exact starting and ending position of structural variants can be challenging. It seems that HiSV could only identify SV at bin pair level (for example 5kb, 10kb). What if one wants to find the exact location of a ~3kb SV?

(2) HiC_breakfinder and EagleC use normal cell lines to construct background models. When normal cell lines are available, can HiSV incorporate this information to provide better identification of SV?

(3) The authors exclude the comparison with EagleC since it is a supervised model. However, EagleC can be pre-trained and applied to 91 Hi-C datasets and 25 HiChIP/ChIA-PET datasets from 105 cancer cell lines or primary tumors. It should be a reasonable comparison to see how the pre-trained EagleC performs on the data used in this manuscript.

(4) The ground truth is defined as SVs with lengths greater than 1MB. Can HiSV predict short-range (<1MB) SVs? How does it compare with other SV detection tools?

I also have one minor comment:

(1) Line 316 dist[(i,j),(p,q)] seems to be a typo. I think you are referring to dist(z_i,j - z_p,q).

Reviewer #3: Li et al present an interesting computational method named HiSV to identify structure variations (SV) from the Hi-C contact map. HiSV out-perform existing unsupervised methods with the ability to detect both intra-chromosomal and inter-chromosomal SVs. Overall, I found the manuscript is reasonably clear and the use of the saliency map is the main novelty of this study. Despite the merit of this method, I have significant concerns about the robustness of the method towards known experimental biases and the choice of free parameters. My detailed comments are listed below.

Major comments:

1. In the introduction section, the authors claim that one advantage of HiSV over HiC_breakfinder and EagleC is that HiSV does not need a background model such that HiSV can be applied to tissue samples. First, this statement is not accurate. EagleC only requires a handful of negative samples from the normal cell lines for training. The authors of EagleC actually apply their method to over 100 Hi-C datasets from either cancer cell lines or primary tumors. HiNT-TL requires a background model, however, a generic background model averaged from multiple cell types also works well. On the other hand, the usage of the background model is helpful to distinguish SVs from 3D genome features such as interactions between A/B compartments, especially SVs within a short distance (e.g., less than 1Mb). I wonder what is the lower bound of the range of SVs that HiSV can identify, because long-range intra-chromosomal SVs are similar to inter-chromosomal SVs while short-range intra-chromosomal SVs are much harder to identify. The authors should provide a comprehensive assessment of the capability of HiSV in detecting various size ranges of intra-chromosomal SVs. For example, the authors should add more groups (e.g., < 1Mb) in Figure 3D and add a category of SVs that were missed by HiSV but appeared in the ground truth. Simulation results would also be quite helpful to demonstrate the capability of HiSV.

2. The authors should provide more explanation on how to choose parameter t. For example, given what kind of sequence depth, should we choose 0.5 rather than 0.6? The authors may consider down-sampling a Hi-C dataset and run HiSV under different values of t and assess what would be the best choice of t given different sequencing depths. In addition, it is unclear if parameter t is sensitive to the choice of bin size.

3. The authors should take Hi-C biases such as GC content, the number of restriction sites, and mappability into account when calculate the average profiles, which is used to annotate SVs into different types.

Minor comments:

1. The authors should add a description of the output file in the GitHub repo

2. The authors should explain the definition of q_values in the caption of Figure 1C

3. Line 73, typo "specie samples"

**Have the authors made all data and (if applicable) computational code underlying the findings in their manuscript fully available?**

Reviewer #1: Yes

Reviewer #2: Yes

Reviewer #3: Yes

PLOS authors have the option to publish the peer review history of their article (what does this mean?). If published, this will include your full peer review and any attached files.

Reviewer #1: No

Reviewer #2: No

Reviewer #3: No
---

## [Decision Letter · Decision Letter 1]

24 Nov 2022

Dear Dr. Gao,

We are pleased to inform you that your manuscript 'HiSV: a control-free method for structural variation detection from Hi-C data' has been provisionally accepted for publication in PLOS Computational Biology.

Before your manuscript can be formally accepted you will need to address Reviewer #3's additional suggestion on the text and also complete some formatting changes, which you will receive in a follow up email. A member of our team will be in touch with a set of requests.

Best regards,

Jie Liu

Academic Editor

PLOS Computational Biology

Jian Ma

Section Editor

PLOS Computational Biology

If possible, please address the point from Reviewer 3.

Reviewer's Responses to Questions

**Comments to the Authors:**

Reviewer #1: In this revised version, the authors add more experiments and made efforts in addressing my concerns. I think the work is improved and I am satisfied with the current version.

Reviewer #2: The authors have addressed all my concerns.

Reviewer #3: The authors have addressed most of my concerns in the revised manuscript. There is only one place where I think the revised text is not accurate. In the revised introduction, the authors stated that "These methods also cannot accurately predict the SVs of other species because of the variation in 3D genome organization features between species [14].” This statement is not true. The authors of EagleC tested their method in mouse samples. Additionally, the authors of HiSV did not show any results on applying HiSV in other species. I would suggest authors remove this sentence or revise it.

Regarding the comparison with EagleC, as also pointed out by reviewer 1, it is reasonable to assess the performance of EagleC in datasets tested by HiSV. The authors indeed compare these two methods in HCC1954 (the only sample not used as training datasets in EagleC). The results in the response show that EagleC is better than HiSV in detecting intra-chromosomal especially short ones (< 1Mb) while HiSV has better performance than EagleC in detecting inter-chromosomal SVs. I wonder if the trend is consistent when the authors apply HiSV in other samples tested by EagleC (e.g., MCF-7). This may be beyond the scope of this paper but it could significantly strengthen the statement of HiSV in accurately detecting large-scale/inter-chromosomal SVs if similar results can be observed in other cancer cell lines or patient samples.

**Have the authors made all data and (if applicable) computational code underlying the findings in their manuscript fully available?**

Reviewer #1: Yes

Reviewer #2: Yes

Reviewer #3: Yes

PLOS authors have the option to publish the peer review history of their article (what does this mean?). If published, this will include your full peer review and any attached files.

Reviewer #1: No

Reviewer #2: No

Reviewer #3: No

---

## [Editor Report · Acceptance letter]

3 Jan 2023

PCOMPBIOL-D-22-01245R1 

HiSV: a control-free method for structural variation detection from Hi-C data

Dear Dr Gao,

I am pleased to inform you that your manuscript has been formally accepted for publication in PLOS Computational Biology. Your manuscript is now with our production department and you will be notified of the publication date in due course.

With kind regards,

Bernadett Koltai
